# Adaptation of the late ISC pathway in the anaerobic mitochondrial organelles of *Giardia intestinalis*

**Alžběta Motyčková**[1], **Luboš Voleman**[1], **Vladimíra Najdrová**[1], **Lenka Arbonová**[1], **Martin Benda**[1], **Vít Dohnálek**[1], **Natalia Janowicz**[1], **Ronald Malych**[1], **Róbert Šuťák**[1], **Thijs J. G. Ettema**[2], **Staffan Svärd**[3], **Courtney W. Stairs**[4]*, **Pavel Doležal**[1]*

1 Department of Parasitology, Faculty of Science, Charles University, BIOCEV, Průmyslová Vestec, Czech Republic, 2 Laboratory of Microbiology, Wageningen University and Research, Wageningen, The Netherlands, 3 Department of Cell and Molecular Biology, Biomedical Center (BMC), Uppsala University, Uppsala, Sweden, 4 Department of Biology, Lund University, Lund, Sweden

* courtney.stairs@biol.lu.se (CWS); pavel.dolezal@natur.cuni.cz (PD)

**Data Availability Statement:** Relevant data are within the manuscript and its Supporting Information files. Alignments and tree files are available at figshare (https://figshare.com/s/

## Abstract

Mitochondrial metabolism is entirely dependent on the biosynthesis of the [4Fe-4S] clusters, which are part of the subunits of the respiratory chain. The mitochondrial late ISC pathway mediates the formation of these clusters from simpler [2Fe-2S] molecules and transfers them to client proteins. Here, we characterized the late ISC pathway in one of the simplest mitochondria, mitosomes, of the anaerobic protist *Giardia intestinalis* that lost the respiratory chain and other hallmarks of mitochondria. In addition to IscA2, Nfu1 and Grx5 we identified a novel BolA1 homologue in *G. intestinalis* mitosomes. It specifically interacts with Grx5 and according to the high-affinity pulldown also with other core mitosomal components. Using CRISPR/Cas9 we were able to establish full *bolA1* knock out, the first cell line lacking a mitosomal protein. Despite the ISC pathway being the only metabolic role of the mitosome no significant changes in the mitosome biology could be observed as neither the number of the mitosomes or their capability to form [2Fe-2S] clusters *in vitro* was affected. We failed to identify natural client proteins that would require the [2Fe-2S] or [4Fe-4S] cluster within the mitosomes, with the exception of [2Fe-2S] ferredoxin, which is itself part of the ISC pathway. The overall uptake of iron into the cellular proteins remained unchanged as also observed for the *grx5* knock out cell line. The pull-downs of all late ISC components were used to build the interactome of the pathway showing specific position of IscA2 due to its interaction with the outer mitosomal membrane proteins. Finally, the comparative analysis across Metamonada species suggested that the adaptation of the late ISC pathway identified in *G. intestinalis* occurred early in the evolution of this supergroup of eukaryotes.

## Author summary

Anaerobic parasitic protists, such as *Giardia intestinalis*, have dramatically reduced their mitochondrial metabolism. Large mitochondrial networks known from aerobic

---

8fbd1368814dbd11192c, DOI:10.6084/m9.
figshare.19772155).

**Funding:** This work was supported by the Czech
Science Foundation grant (20-25417S) and the
European Regional Development Fund 'Centre for
research of pathogenicity and virulence of
parasites' (No. CZ.02.1.01/0.0/0.0/16_019/
0000759) ()https://ec.europa.eu/regional_policy/
en/funding/erdf/ to PD, the grant from the Charles
University Grant Agency (project number 1396217)
to AM (www.cuni.cz), the European Molecular
Biology Organization long-term fellowship (ALTF-
997-2015) (https://www.embo.org/) and Swedish
Research Council (Vetenskapsrådet starting grant
2020-05071) (https://www.vr.se/english.htm) to
CWS. The Imaging Methods Core Facility at
BIOCEV were supported by the grant from MEYS
CR (Large RI Project LM2018129 Czech-
BioImaging) (www.msmt.cz) and the European
Regional Development Fund (project No.
CZ.02.1.01/0.0/0.0/18_046/0016045) (https://ec.
europa.eu/regional_policy/en/funding/erdf/). The
funders had no role in study design, data collection
and analysis, decision to publish, or preparation of
the manuscript.

**Competing interests:** The authors have declared
that no competing interests exist.

eukaryotes have evolved into minimalist vesicles that are still enclosed by a double membrane but have lost the mitochondrial genome, respiratory chain, and most mitochondrial enzymes. However, these small mitochondria still contain the so-called ISC pathway, which synthesizes iron-sulfur clusters, essential cofactors for the function of respiratory chain complexes, and other mitochondrial enzymes. In this work, we have begun to characterize the ISC pathway in *G. intestinalis* to understand whether and why iron-sulfur clusters are formed in these small mitochondrial organelles known as mitosomes. We found that while the 'early' part of the ISC pathway responsible for the formation of the simplest [2Fe-2S] clusters is functional in mitosomes, the function of the 'late' components involved in [4Fe-4S] cluster formation remains unclear due to unknown substrates that require these clusters. Identifying the role of the "late" ISC pathway is crucial to understanding why mitosomes remain present in anaerobic eukaryotes at all.

## Introduction

*Giardia intestinalis* is an anaerobic parasitic protist that lives in the epithelium of the small intestine of mammals, where it causes giardiasis [1]. It belongs to the Metamonada supergroup of eukaryotes that is comprised of organisms that contain mitochondria-related organelles (MRO) that lack organellar genomes and cristae and that are adapted to life with little or no oxygen [2]. The so-called 'mitosomes' of *G. intestinalis* are one of the simplest MROs known among eukaryotes, as they contain only a single metabolic pathway, the iron-sulfur (Fe-S) cluster synthesis (ISC) [3–5].

Fe-S clusters function as cofactors of proteins (Fe-S proteins) in all living organism. In eukaryotes, they participate in essential biological processes in various compartments such as DNA maintenance in the nucleus, electron transport chains in mitochondria, and protein translation in the cytoplasm [6–8]. In humans, about 70 different Fe-S proteins have been identified [7].

In aerobic eukaryotes, the formation of Fe-S clusters for all cellular proteins begins in mitochondria via the activity of the ISC pathway, which can be functionally divided into the early or late acting complex of proteins [9]. In 'classical' mitochondria (Fig 1A), the early ISC pathway produces [2Fe-2S] clusters on the scaffold protein IscU [10] via the activity of a complex consisting of cysteine desulfurase IscS [11], its accessory subunit Isd11 [12–14] and an acyl carrier protein [15–17]. The actual transfer of sulfur to IscU is facilitated by frataxin [18] and the electrons for cluster formation are provided by reduced ferredoxin (Fdx), which itself is a [2Fe-2S] protein [19]. However, the source of iron and the mechanism of iron transfer to the cluster remain elusive. Upon the formation of [2Fe-2S] cluster on IscU, a chaperone complex consisting of Hsp70 and HscB transfers the cluster to glutaredoxin 5 (Grx5) apoprotein [20].

Grx5 acts as the central dividing point between the early and late ISC pathway at which the assembled [2Fe-2S] cluster is either (i) transferred to the target mitochondrial [2Fe-2S] apoproteins, (ii) exported to the cytosol as an enigmatic X-S compound or (iii) enters the late ISC machinery [9,21]. The late ISC machinery starts with the transfer of two [2Fe-2S] clusters from Grx5 to a complex of IscA1, IscA2 and Iba57 [22] where the [4Fe-4S] cluster is formed [23]. The newly created [4Fe-4S] clusters are delivered to apoproteins with the help of Nfu1 [24,25] and Ind1, the latter being specifically involved in [4Fe-4S] cluster-binding for the complex I assembly [26]. Recently, two conserved factors BolA1 and BolA3 have been shown to participate in the transfer of [4Fe-4S] clusters to apoproteins in mitochondria [27]. BolA1 and BolA3 have overlapping functions, but preferentially act on Grx5 and Nfu1, respectively [25].

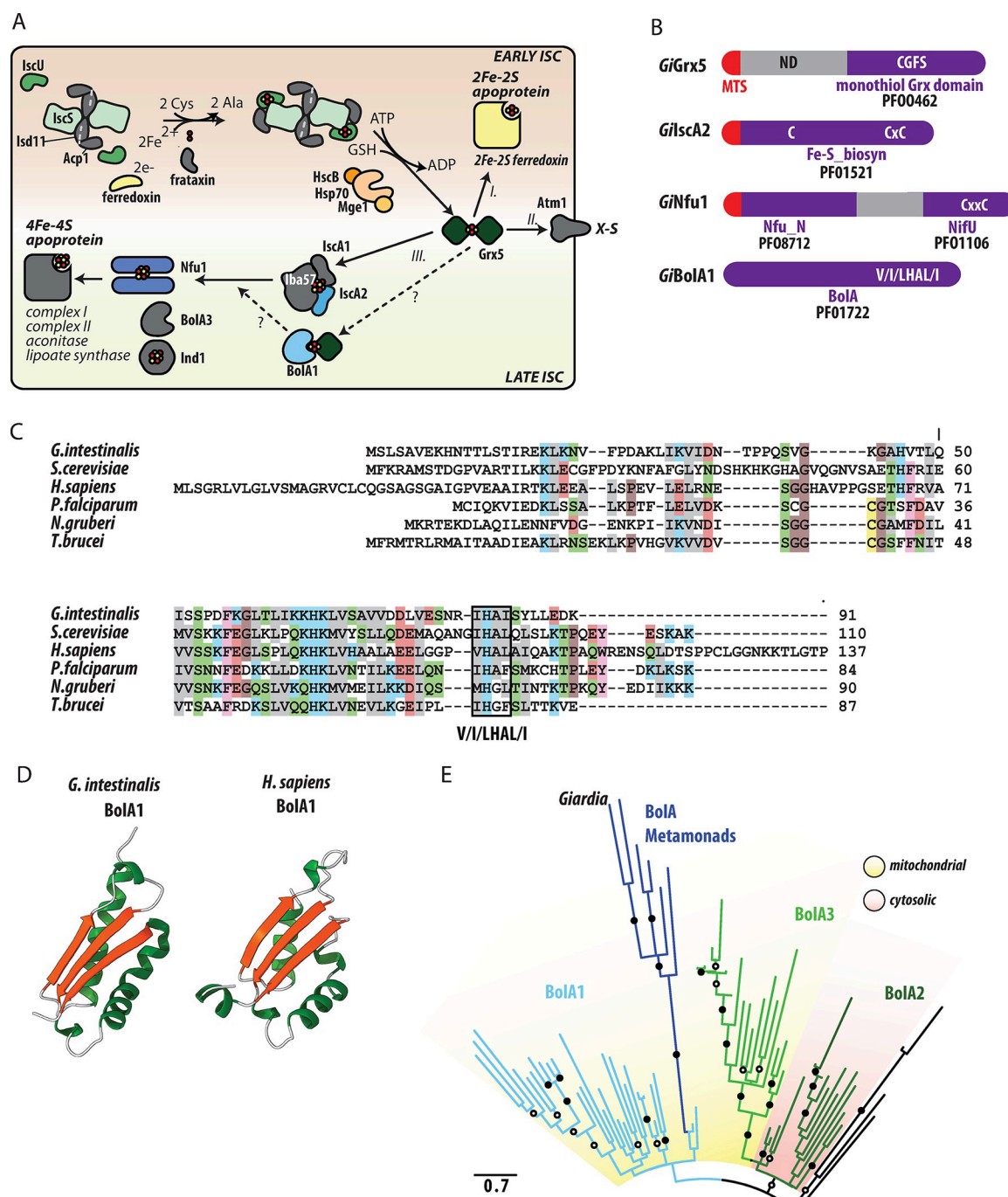

**Fig 1. *G. intestinalis* encodes components of the late ISC pathway.** (A) Schematic representation of mitosomal ISC pathway against the current model of the pathway as it occurs in aerobic mitochondria of fungi and animals. The early ISC pathway starts with a complex containing cysteine desulfurase IscS and its activator Isd11 with acylated ACP1. The complex is bound by IscU, on which the [2Fe-2S] cluster is built. Sulfur is released from cysteine by IscS and its transfer to the cluster is facilitated by frataxin, and the electrons for cluster formation are provided by reduced ferredoxin. Upon the formation of [2Fe-2S] cluster on IscU, a chaperone complex consisting of Hsp70 and HscB transfers the cluster to glutaredoxin 5 (Grx5). From Grx5, [2Fe-2S] cluster is either (*I.*) transferred to the target mitochondrial [2Fe-2S] apoproteins, (*II.*) exported to the cytosol as an enigmatic X-S compound or (*III.*) enters the late ISC machinery. The late ISC machinery starts with the transfer of two [2Fe-2S] clusters from Grx5 to a complex of IscA1, IscA2, and Iba57 where the [4Fe-4S] cluster is formed. The newly formed [4Fe-4S] clusters are delivered to apoproteins with the help of Nfu1 and Ind1. The precise role of BolA proteins remains unknown, but BolA1 was shown to interact with Grx5, while BolA3 interacts with Nfu1. The mitochondrial components that are missing in *Giardia* mitosomes are shown in gray. The early and late ISC pathways are distinguished by the background color. (B) Domain structure of *Gi*Grx5, *Gi*IscA2, *Gi*Nfu1, and *Gi*BolA1. The respective sequence motifs and Pfam

accession numbers are shown. (C) Protein sequence alignment of the identified *Gi*BolA1 with the homologues from, *Saccharomyces cerevisiae* (Q3E793), *Homo sapiens* (Q9Y3E2), *Plasmodium falciparum* (Q8I3V0), *Naegleria gruberi* (D2V472) and *Trypanosoma brucei* (Q57YM0). BolA signature V/I/LHAL/I motif is highlighted. (D) Structure of *Gi*BolA1 as predicted by AlphaFold2 [75], predicted structure of human BolA1 (*Hs*BolA1) [27] is shown for comparison. (E) Maximum likelihood phylogenetic tree of 70 eukaryotic BolA1 paralogues shows that *Gi*BolA1 and metamonad BolA homologues emerge from within a clade of mitochondrial BolA1 proteins. Summary of bipartition support values (1000 ultrafast bootstraps) greater than 80 or 95 are shown in open and closed circles, respectively.

Importantly, BolA function has previously been associated with aerobic metabolism, which was supported by its absence in anaerobic eukaryotes [28].

It is now generally accepted that the early ISC pathway is a converging evolutionary point of the MROs, *i.e.*, no matter how much the mitochondrion has been modified during evolution, most MROs have retained early ISC components like IscU and IscS [29]. Moreover, some of the MROs like mitosomes of *G. intestinalis* and other anaerobes also contain components of the late ISC pathway [30]. Therefore, here, we sought to experimentally examine the nature of the late ISC pathway in *G. intestinalis*. Using bioinformatics, we identified a BolA1 homologue in *G. intestinalis* mitosomes. It specifically interacts with Grx5 and according to a protein pull-down also with other core mitosomal components. However, the experimental removal of the corresponding genes showed no significant changes in mitosome biology, and the function of the early ISC pathway.

Using enzymatic tagging and series of high/affinity pulldowns, we have generated a robust interactome of the mitosomal late ISC pathway revealing that Grx5, Nfu1 and herein discovered BolA1 orthologue are at the core of the pathway. On the other hand, mitosomal IscA2 appears to function in downstream steps of the pathway. The absence of any known client [4Fe-4S] proteins in the mitosomes suggests that the entire late mitosomal ISC pathway serves only to provide the clusters to non-mitosomal clients.

## Results

The late ISC pathway and the identification of BolA1 in *G. intestinalis*

Previous genomic and proteomic analyses of *G. intestinalis* revealed the presence of three late ISC pathway components; Nfu1, IscA2 and Grx5, hereafter referred to as *Gi*Nfu1, *Gi*IscA2, and *Gi*Grx5, respectively (Fig 1A). All three proteins possess highly conserved cysteine residues that are necessary for the coordination of the Fe-S cluster. *Gi*Grx5 contains the CGFS motif of monothiol glutaredoxins (Figs 1B and S1A), the C-terminal domain of *Gi*Nfu1 carries a CxxC motif (Figs 1B and S1B) and *Gi*IscA2 carries a $Cx_nCxC$ signature motif (Figs 1B and S1C). Both *Gi*Nfu1 and *Gi*IscA2 have a short N-terminal pre-sequence that likely serves as the mitosomal targeting signal. *Gi*Grx5 was previously shown to carry a long non-homologous N-terminal sequence, which is required for targeting but may possibly play an additional role in protein function [31]. Of the two types of IscA proteins known for eukaryotes, only IscA2 was identified in *G. intestinalis* [4].

The presence of these three late ISC components in *G. intestinalis* prompted us to search for other factors that were identified within the late pathway. Specifically, the orthologues of BolA, Iba57 and Ind1 proteins were searched using hidden Markov model (HMM) profiles against the *G. intestinalis* genome. Interestingly, while the last two searches did not result in the identification of positive hits, a single BolA orthologue was found in *G. intestinalis* (*Gi*BolA1) (Fig 1B and 1C). The protein could be readily identified in the conceptual proteomes of all genotypes (assemblages) including new genome assembly of WBc6 [32] but was missing from the original reference genome, probably due to its small size [33]. The amino acid sequence of *Gi*BolA1 contains signature V/I/LHAL/I motif towards the C-terminus [34]

but no putative N-terminal targeting sequence, as is common to most other BolA proteins (e.g., Fig 1C). Structural prediction of *Gi*BolA1 using AlphaFold 2 revealed an αβαβ topology that matches experimentally solved or predicted structures of BolA homologs from both eukaryotes and prokaryotes (Fig 1D) [35,36]. The only structural difference is a short C- terminal α-helix missing in *Gi*BolA1 (Fig 1D). Given the occurrence of three BolA proteins in eukaryotes, phylogenetic analysis was performed to determine which of three eukaryotic BolA paralogues, functioning in the cytosol (BolA2) [37] or mitochondria (BolA1 and BolA3) [28,38] is present in *G. intestinalis*. The analysis showed that *Gi*BolA1 and other BolA proteins that could be identified in the Metamonada supergroup emerge from within a clade of BolA1 proteins (Fig 1E) suggesting that *G. intestinalis* contains an orthologue of mitochondrial BolA1, which would hence be expected to be localized in mitosomes.

## *Gi*BolA1 and other late ISC pathway components are mitosomal proteins

To test whether *Gi*BolA1 is indeed a mitosomal component, the protein was expressed with the C-terminal biotin acceptor peptide tag (BAP) tag. Immunodetection of the tag by fluorescence microscopy showed clear colocalization of *Gi*BolA1 with the mitosomal marker GL50803_9296 [3] (Fig 2A). Western blot analysis of the cellular fractions revealed specific presence of the protein in the high-speed pellet (HSP) fraction that is enriched for mitosomes (Fig 2B). Except for *Gi*Grx5 [31], the mitosomal localization of other late ISC components had not been previously experimentally confirmed. Therefore, analogously, all three proteins were expressed with the C-terminal BAP tag and their cellular localization was detected in the fixed cells (Fig 2A) and in the cell fractions (Figs 2B and S2). All proteins specifically localized in the mitosomes. Furthermore, we tested whether BAP-tagged proteins are within the mitosomes or are accumulated on the surface of the organelle as a possible result of protein overexpression. To this end, a protease protection assay was performed on *G. intestinalis* expressing BAP-tagged proteins whereby HSPs were incubated with trypsin in presence or absence of a membrane-solubilizing detergent. Proteins encased by one or more membranes will be inaccessible to trypsin and will therefore be detected by standard immunoblotting in the absence but not presence of the detergent (Fig 2C). Unlike the outer membrane marker *Gi*Tom40, all late ISC components were resistant to protease treatment as the mitosomal matrix marker *Gi*IscU. As a control, mitosomal membrane solubilization resulted in overall protein degradation. In summary, all four proteins were found specifically located within mitosomes, suggesting that the minimalist late ISC pathway occurs within the organelles.

## Mitosomal BolA1 specifically interacts with Grx5 and other mitosomal ISC components

Recent studies on human BolA1 proteins showed a specific interaction of mitochondrial BolA1 with Grx5 during the stabilization of [2Fe-2S] cluster on Grx5 [27]. Using a yeast two hybrid (Y2H) assay, we tested whether mitosomal BolA1 also interacts with Grx5. Indeed, the assay was able to show the interaction between *Gi*BolA1 and *Gi*Grx5 (Fig 2D). Previous studies in yeast identified the specific residues of BolA and Grx5 critical for interaction [27]. Therefore, we tested whether the same molecular interaction can also be demonstrated for the *Giardia* proteins. Specifically, the cysteine residue (position 128) within the CGFS motif of *Gi*Grx5 and a highly conserved histidine residue (position 82) of *Gi*BolA1, that were both shown to coordinate Fe-S cluster [39]. In both cases, the introduced mutations abolished the positive interaction in Y2H (Fig 2D). These results strongly suggest that the mechanism of interaction is conserved for the late ISC components in the *G. intestinalis* mitosomes. However, the

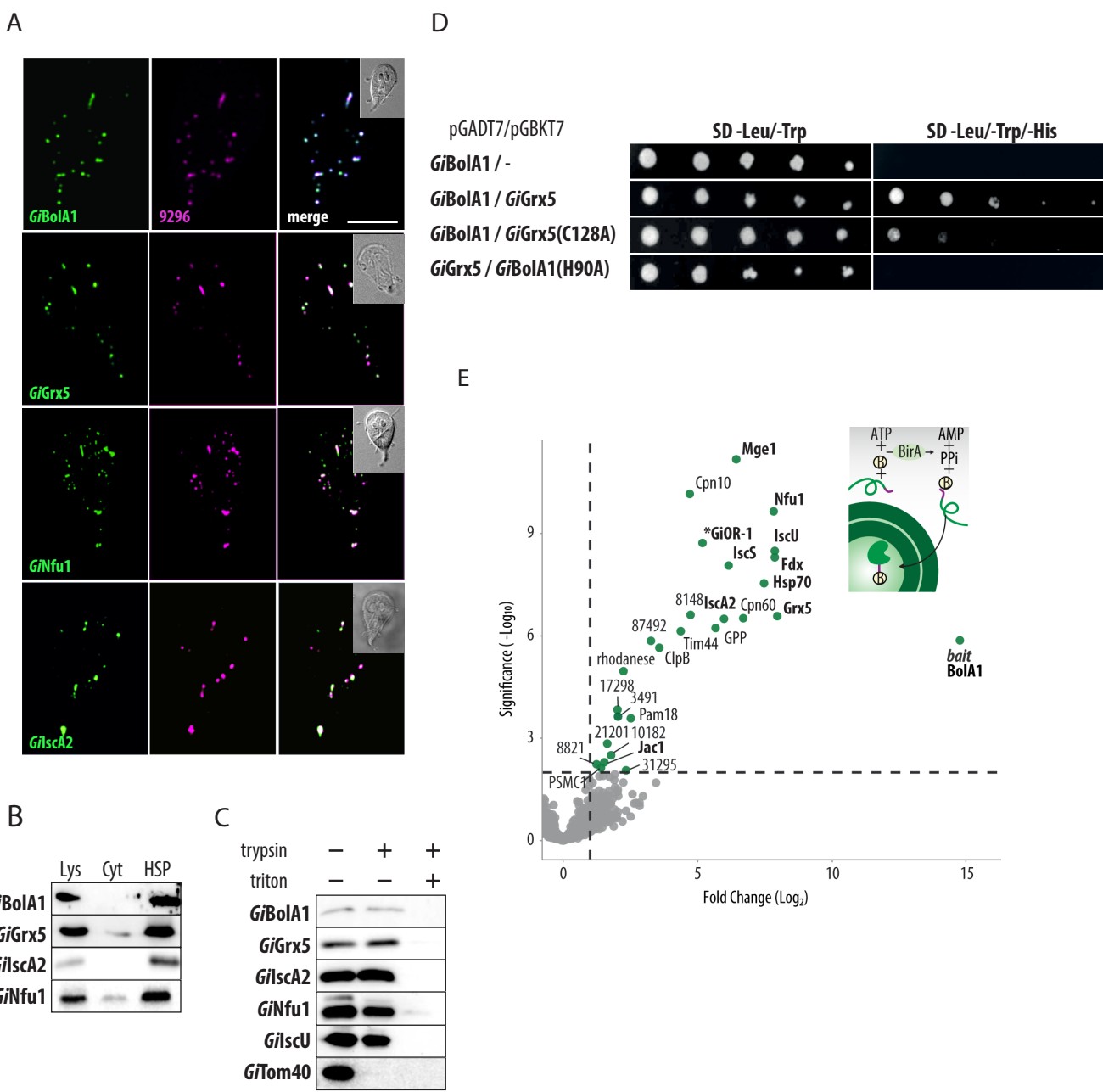

**Fig 2. *Gi*BolA1 is a mitosomal protein that specifically interacts with *Gi*Grx5 and other ISC components.** (A) BAP-tagged *Gi*BolA1, *Gi*Grx5, *Gi*Nfu1 and *Gi*IscA2 were expressed in *G. intestinalis* and the proteins were detected by anti-BAP antibody (green). The co-colocalization with mitosomal marker GL50803_9296 (magenta) is shown. The DIC image of the cell is shown in the inlet, the scale bar represents 5 μm. (B) Detection of BAP-tagged *Gi*BolA1, *Gi*Grx5, *Gi*Nfu1 and *Gi*IscA2 in cellular fractions, lys–cell lysate, cyt—cytosol, HSP–high speed pellet fraction. (C) Protease protection assay of late ISC components and the markers of the outer mitosomal membrane (*Gi*Tom40) and the mitosomal matrix (*Gi*IscU). High-speed pellets isolated from *G. intestinalis* expressing BAP-tagged *Gi*BolA1, *Gi*Grx5 *Gi*IscA2 and *Gi*Nfu1 were incubated with 20 μg/ml trypsin and 0.1% Triton X-100. The samples were immunolabeled with antibodies against the BAP tag, *Gi*Tom40 and *Gi*IscU. (D) Serial dilutions of Y2H assay testing the protein interactions between *Gi*BolA1 and *Gi*Grx5. The introduction of specific mutations of conserved residues (H90A *Gi*BolA1 and C128A *Gi*Grx5) abolished the interaction, double and triple dropout medium was used to test the presence of the plasmids and the interaction of the encoded proteins, respectively. (E) Affinity purification of the *in vivo* biotinylated *Gi*BolA1 with the DSP-crosslinked interacting partners. (top right) Scheme of the *in vivo* biotinylation of the C-terminal BAP-tag of *Gi*BolA1 by cytosolic BirA. (left) Volcano plot of the statistically significant hits obtained from the protein purification on streptavidin coupled Dynabeads. Components involved in ISC pathway are shown in bold letters.

analogous assay did not show any interaction between *Gi*BolA1 and *Gi*Nfu1 (S3 Fig), that would be expected if *G. intestinalis* BolA represented a BolA3 homologue [25].

To reveal the complex *in vivo* interactions of *Gi*BolA1, we used a previously established method of enzymatic tagging in *G. intestinalis* that is based on co-expression of the biotin ligase (BirA) and protein of interest tagged by BAP [3]. In the presence of ATP, BirA specifically biotinylates the lysine residue within the BAP tag. Therefore, a BAP-tagged *Gi*BolA1 was introduced into *G. intestinalis* expressing cytosolic BirA. The mitosomes-enriched HSP was incubated with the chemical crosslinker DSP and *Gi*BolA1-BAP was purified on streptavidin-coupled magnetic beads (see Materials and Methods for more details). The purified cross-linked complexes were subjected to proteomic analysis and the resulting peptide mass spectra were searched against the predicted proteome of *G. intestinalis* [40]. Data obtained from the biological and technical triplicates (S1 Table) were displayed in a volcano plot showing the fold change of protein abundance compared to the negative control (Fig 2E). In total, 26 significantly-enriched proteins were identified. *Gi*Grx5 represented the most enriched interactor but other ISC components (Nfu1, IscA2, Fdx, IscU, IscS, Hsp70, Jac1) also appeared among the most significant enriched proteins (Fig 2E). The remaining proteins represented mitosomal proteins involved in protein import and folding, and mitosomal proteins of unknown function. At least one probable non-mitosomal protein (PSMC1, Proteasome 26S Subunit, ATPase 1 homologue) was identified among the significantly enriched proteins (Fig 2E and S1 Table) suggesting minimal contamination from non-mitosomal proteins in this pulldown method. The dominant presence of mitosomal matrix proteins in the presented interactome strongly suggests that *Gi*BolA1 is localized in the mitosomal matrix.

## Knockout of the *bolA1* gene does not affect mitosomal iron-sulfur cluster assembly or the incorporation of iron into proteins

BolA was previously thought to be restricted to aerobic eukaryotes [28], therefore all functional analyses have been performed on aerobic model organisms [41]. Having established the integration of *Gi*BolA1 within the mitosomal late ISC pathway, we next examined the role of *Gi*BolA1 in the formation of Fe-S clusters. To this end, using the recently established CRISPR/Cas9-mediated gene knockout approach [42] *G. intestinalis* cell line lacking the *bolA1* gene (ΔbolA1) was generated (Fig 3A and 3B). The gene knockout was verified by PCR in gDNA for the absence of the the *bolA1* gene and the presence of a a homologous recombination cassette (HRC) (Fig 3A). Furthermore, no *bolA1* mRNA was detected in cDNA prepared from cells (Fig 3B). Finally, proteomic analysis of the HSP fraction enriched in mitosomes showed the absence of *Gi*BolA1 compared to the control fraction (S3 Table).

To assess the mitosome-related phenotype in Δ*bolA1* cells, we checked if the number of organelles per cell changed compared to the control cells. Both cell lines showed a comparable number of mitosomes (Figs 3C and S4).

Given the absence of typical client proteins from the mitochondrial ISC pathway, such as proteins involved in the electron transport chain, the TCA cycle and cofactor biosynthesis [43–46], we performed an unbiased search for Fe-S proteins within the conceptual *G. intestinalis* cellular proteome using MetalPredator [47]. Upon manual checking with available literature and structural information, 40 proteins were identified that bind [4Fe-4S] clusters (Fig 3D and S2 Table). Of these, 19 were predicted to function in the cytosol in energy, redox, amino acid, and nitrogen metabolism, as well as cofactor biosynthesis and protein translation. There were 11 nuclear proteins identified, participating either in DNA or RNA metabolism. The remaining components corresponded to transient cluster carriers of the mitosomal ISC machinery and the cytosolic iron–sulfur assembly (CIA) pathway [48]. The only mitosomal

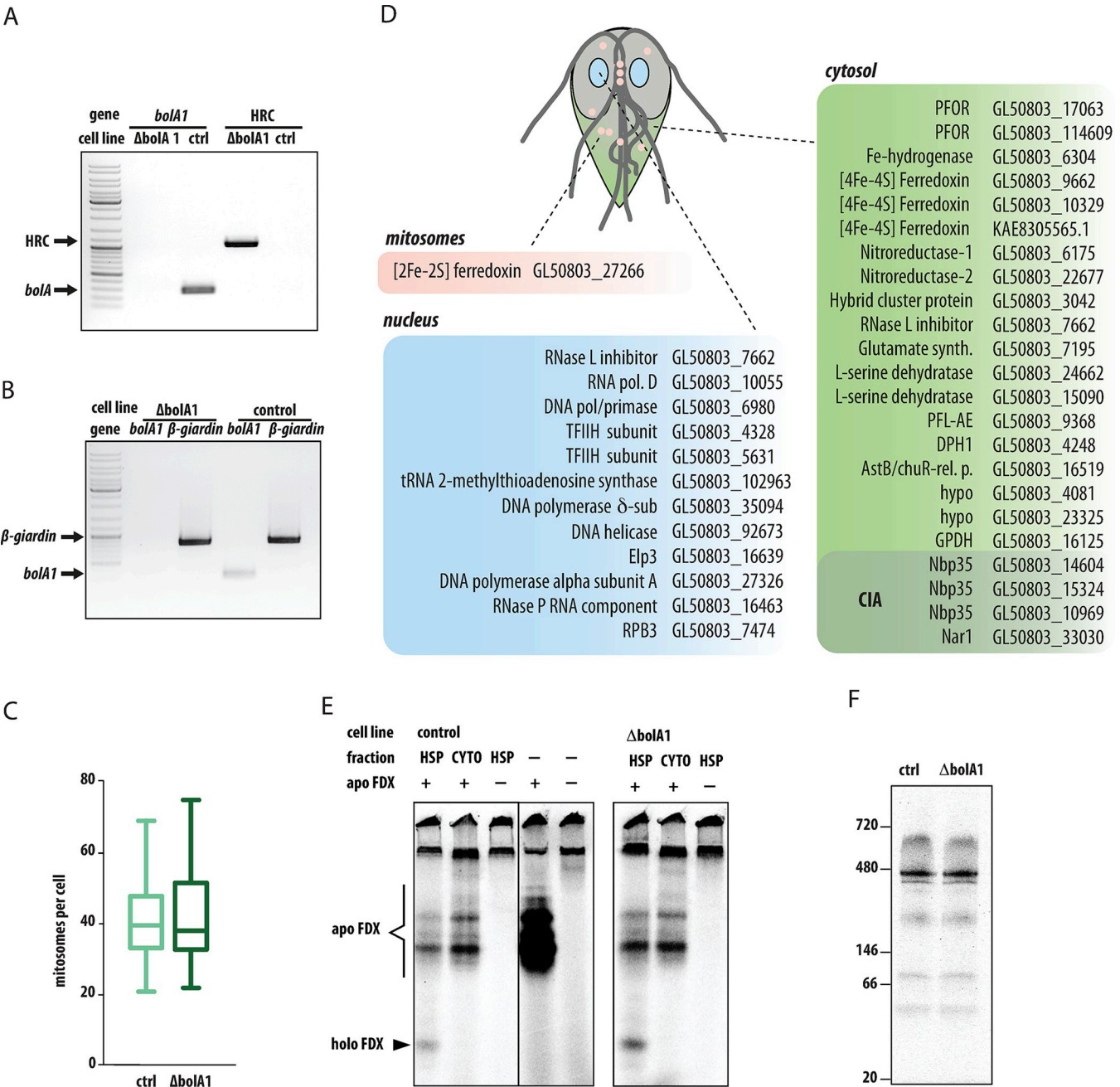

**Fig 3. Characterization of BolA1 knockout (ΔbolA1) cell line.** (A) The ΔbolA1 cell line was tested for the presence of *bolA1* gene and the integration of homologous recombination cassette (HRC) by PCR on gDNA, (B) the expression of *bolA1* gene in ΔbolA1 cell line was tested by PCR on the cDNA, *β-giardin* was used as a control gene. (C) The number of mitosomes per cells in Cas9-expressing (n = 64) and ΔbolA1 cells (n = 107), the error bars of the box plot depict min to max values. (D) The list of predicted 40 Fe-S proteins in *G. intestinalis* includes only one mitosomal protein, [2Fe-2S] ferredoxin, that itself participates in the ISC pathway. All putative clients that require [4Fe-4S] clusters are localized in the cytosol or in the nucleus (S2 Table). (E) In vitro Fe-S cluster assembly assay in the mitosome-enriched high-speed pellet (HSP) fraction. The organelles or the cytosolic fraction were incubated in the reaction buffer supplemented with apoferredoxin (apo FDX), $^{35}$S-labeled L-cystein and ferrous ascorbate at 25°C for 60 min. The incorporation of $^{35}$S and the assembly of holoferreoxin (holo FDX) was analyzed on nondenaruring protein gel. The analysis of HSP from ΔbolA1 cell line showed that the activity was not abolished in the absence BolA1. (F) Incorporation of $^{55}$Fe to *G. intestinalis* proteins after 72 h incubation with radioactive iron isotope in the form of ferric citrate. Comparisons of control and ΔbolA1 cell extracts show comparable levels of iron incorporation.

protein with a stable Fe-S cluster is [2Fe-2S] ferredoxin, which is itself directly involved in the ISC pathway as an electron carrier. Of course, we cannot rule out the presence of a previously unknown protein with a unique cluster binding domain/motif in mitosomes, but the present data suggest that mitosomes lack any client [4Fe-4S] protein for their late ISC pathway.

Given the absence of the client [4Fe-4S] proteins, we investigated if the ΔbolA1 cells retained functional early ISC pathway that could be affected by the defect in the assembly of the cluster on [2Fe-2S] ferredoxin. To this end, we employed a modified *in vitro* assay previously used to monitor the activity of Fe-S cluster assembly in *G. intestinalis* and *T. vaginalis* [5,49]. Briefly, the isolated organellar or cytosolic fractions of *G. intestinalis* were incubated with an apoprotein, (apoferredoxin from *T. vaginalis*), ferrous ascorbate, and $^{35}$S-labeled L-cysteine. In this work, however, we omitted the addition of DTT that functions as a strong reductant masking the role of endogenous [2Fe-2S] ferredoxin as an electron donor in the reaction [50,51] (see Methods section for more details). Upon 60 min incubation the reaction was resolved on a nondenaturing gel to preserve the cluster on the holoprotein, the gel was dried and exposed to a storage phosphor screen. Only reactions whereby the apoprotein was incubated with the HSP fraction from control or ΔbolA1 cells produced holoferredoxin form (Fig 3E). Neither the cytosolic fraction nor the reaction buffer alone enabled the formation of the holoprotein and $^{35}$S-specific bands corresponded to sulfur only incorporation into the apoprotein or other proteins in the cellular fraction (Fig 3E). Collectively this suggests that the removal of *Gi*BolA1 from mitosomes does not affect the availability of endogenous [2Fe-2S] ferredoxin for the early ISC pathway.

We further investigated whether the absence of *Gi*BolA1 affects the overall cellular iron uptake and its incorporation into proteins as defects in iron metabolism, including the accumulation of iron, were observed for the yeast mutants lacking BolA1 or its Grx5 partner [52,53]. To this end, control and ΔbolA1 cells were grown in culture medium supplemented with 0.5μM $^{55}$Fe for 72 hours. After washing steps and cellular lysis by sonication, the samples were resolved on blue native PAGE, and the dried gel was exposed to the storage phosphor screen. Several dominant and minor bands were specifically labeled by $^{55}$Fe without knowing their identity or whether they represent Fe-S or another iron-binding protein. However, no differences were detected between the control and ΔbolA1 cells, and therefore iron incorporation at least at the level of the cell lysate was not affected by the absence of *Gi*BolA1.

Analogous data were obtained when the experiment was performed with cells lacking the BolA1 partner protein, *Gi*Grx5, which was also generated by CRISPR/Cas9 (S5 Fig).

## Interactome of late ISC components reveals a downstream role of IscA2

Characterization of late ISC pathway in mitochondria has relied largely on genetic and biochemical approaches *e.g.*, [25–27,53–55]. Here, we chose to continue with the affinity-purification proteomics, which to our knowledge has not yet been used in this context, to characterize the pathway in *G. intestinalis* mitosomes. The combination of protein specific interactomes as the one obtained above for *Gi*BolA1 can yield a spatial reconstruction of the pathway [56]. To this aim, proteins co-purified in complexes chemically crosslinked to *Gi*Grx5, *Gi*Nfu1, and *Gi*IscA2 were identified by mass spectrometry. The returned datasets contained 47, 30, and 22 statistically significant proteins of three independent sets of experiments, respectively (Fig 4A-C and S1 Table).

The final combined dataset which also included the *Gi*BolA1 pulldown data was plotted in a heat map using log2 transformed fold difference values (Fig 4D). Hierarchical clustering showed a close relationship between the *Gi*BolA1-, *Gi*Nfu1- and *Gi*Grx5-specific protein profiles, while the *Gi*IscA2-specific dataset remained the most distinct. The interactomes of the

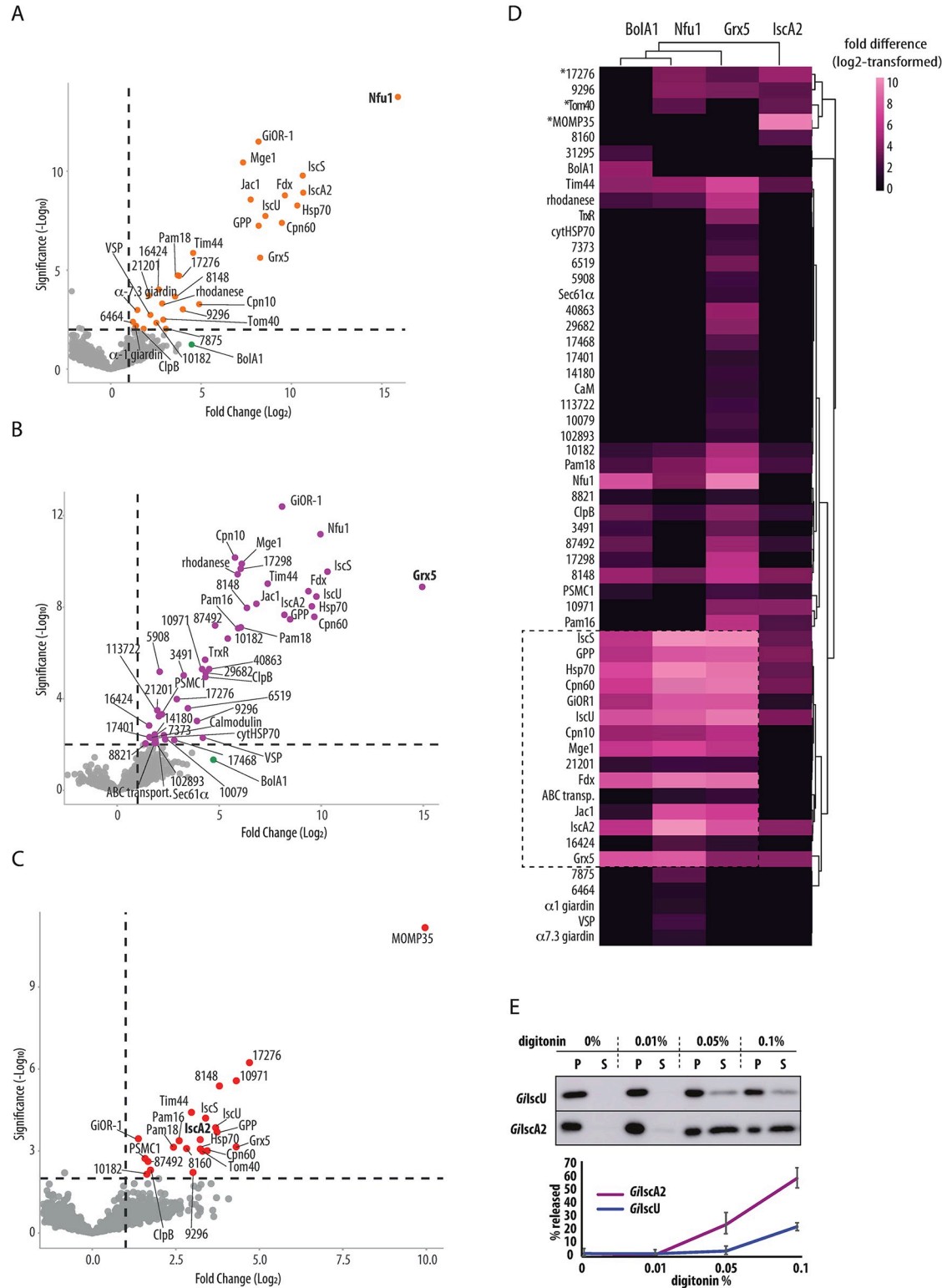

**Fig 4. Proteomic analysis of late ISC pathway.** BAP-tagged *Gi*Grx5, *Gi*Nfu1 and *Gi*IscA2 were *in vivo* biotinylated by cytosolic BirA and purified on streptavidin-coupled Dynabeads upon crosslinking by DSP. (A-C) Volcano plots depict the significantly enriched proteins that co-purified with (A) *Gi*Nfu1, (B) *Gi*Grx5 and (C) *Gi*IscA2. (D) Heatmap of combined significantly enriched proteins for all four late ISC components, (E) Digitonin solubilization of the mitosomes shows differential release of *Gi*IscA2 over *Gi*IscU, P -pellet fraction (retained protein), S–supernatant (released protein). Exemplary western blot of four independent experiments is shown, the error bars show standard deviation.

first three proteins converged over the ISC components, chaperones and the mitosomal processing peptidase (GPP) that corresponds to the 'core' of the mitosomal metabolism (dashed line in Fig 4D). Several low abundance proteins of unknown function (GL50803_21201, GL50803_16424 and ABC transporter GL50803_87446) were also found in the cluster. Interestingly, a thioredoxin reductase (TrxR) homolog (GL50803_9287) was found among several proteins unique to the *Gi*Grx5 dataset (Fig 4B). The protein was previously characterized in *G. intestinalis* as cytosolic protein, yet without any interacting thioredoxin [57]. Our data suggested that TrxR thus could also act in the mitosomes and reduce *Gi*Grx5 to act as a missing reductase system. *Gi*BolA1 was found among enriched proteins in *Gi*Grx5 and *Gi*Nfu1 datasets (Fig 4A and 4B) yet it was not a significant hit due to the incomplete coverage in some of the technical triplicates within biological triplicates. This indicates lower expression levels of *Gi*BolA1 when compared to other late ISC components.

In contrast, the *Gi*IscA2 dataset showed enrichment of the outer mitosomal membrane proteins MOMP35 and GL50803_17276 [3,58]. Additionally, Tom40, a central component of the outer membrane translocase, was identified among the significantly enriched proteins (S1 Table). Unlike the interactomes of the other ISC components, many of the 'core' mitosomal matrix proteins were not significantly enriched in the *Gi*IscA2 interactome. The affinity of *Gi*IscA2 to the outer membrane proteins suggested the possibility that the protein is not localized, at least not completely, in the mitosomal matrix but in the intermembrane space (IMS) or it is associated with the outer mitosomal membrane proteins. The latter could be rejected due to the lack of any transmembrane domains and due to the full protection of *Gi*IscA2 against the externally added protease (Fig 2C). Therefore, the presence of the protein in the IMS was tested. We took advantage of differential sensitivity of the outer and inner mitosomal membranes to digitonin lysis [3,59].

The mitosome-enriched fraction was isolated from cells co-expressing *Gi*IscA2 and the matrix marker *Gi*IscU and incubated with the increasing concentration of digitonin. The release of the proteins from the organelles was monitored via western blot (Fig 4E). Interestingly, *Gi*IscA2 showed a greater proportion of protein released into the supernatant fraction than *Gi*IscU. This may either reflect different physical properties of the proteins or indicate the possibility that *Gi*IscA2 and *Gi*IscU may not be in the same mitosomal subcompartment.

## Adaptation of ISC pathway in Metamonada

Metamonads, with the exception of the secondarily amitochondriate *Monocercomonoides*, host MROs adapted to life without oxygen. According to genomic and transcriptomic analyses, the degree of metabolic reduction of these MROs varies across the Metamonada [30,60]. Some MROs participate in ATP generation and some, such as *G. intestinalis* mitosomes, are involved only in the synthesis of Fe-S clusters. The identification of *Gi*BolA1 prompted us to search the available data for the homologues of BolA and other ISC components in Metamonada.

A BolA homologue was detected in genomes of the parasitic *Giardia muris* and two *Retortamonas* species, and in free-living *Dysnectes brevis*, *Kipferlia bialata* and *Aduncisulcus paluaster* (Fig 5, S4 Table). Similarly to *G. intestinalis*, the vast majority of Metamonada have been found to lack Iba57 and IscA1. The absence of the former correlates with the absence of complex I in these eukaryotes, but both Iba57 and IscA1 are supposed to constitute a complex together with IscA2, on which the [4Fe-4S] cluster is formed [61] This raises the general question whether IscA2, unlike the whole IscA1-IscA2-Iba57 complex, has an indispensable role for anaerobic eukaryotes. Analogously, we could not detect the early ISC components Isd11 and ferredoxin reductase (Arh1) in preaxostylids and fornicates (Fig 5). These components were only detected in the less reduced MROs of parabasalids (e.g., *Trichomonas vaginalis*) and in anaeramoebids

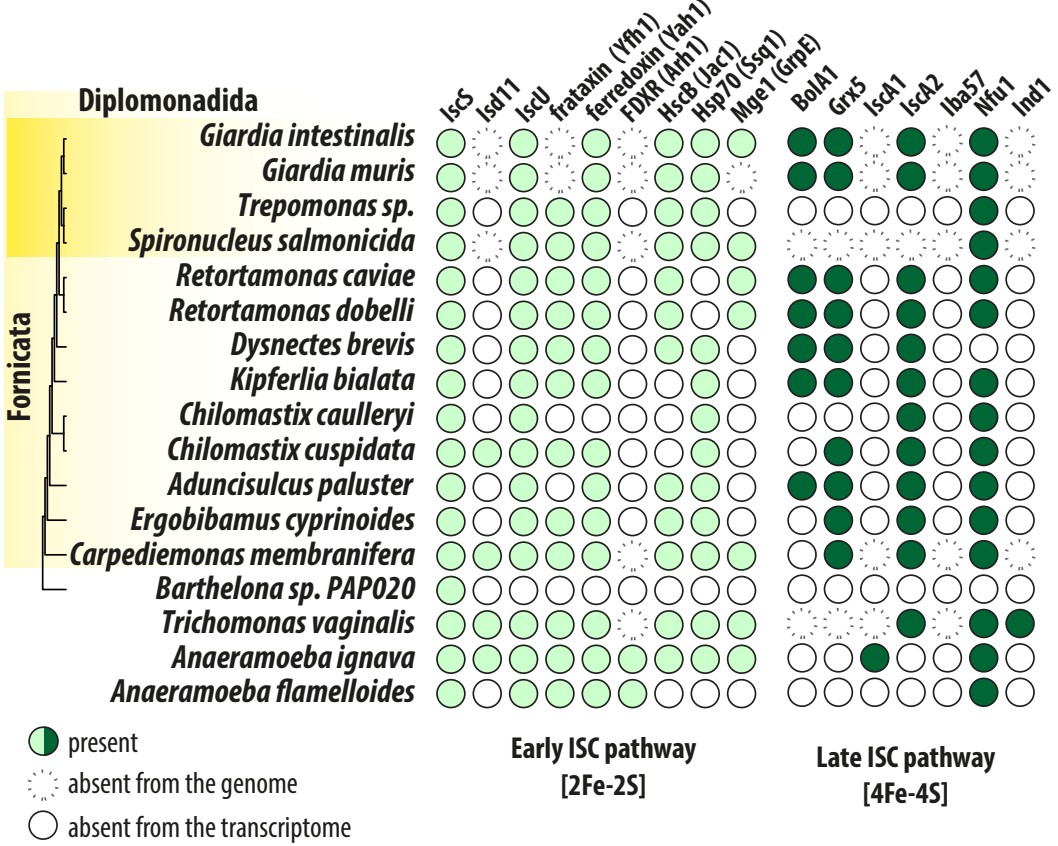

**Fig 5. Adaptation of the ISC pathway in anaerobic mitochondria of Metamonada.** The presence/absence of the ISC components in Metamonada supergroup Absence from the transcriptome means that only transcriptomic data were available for the analysis.

[60]. Of course, additional components can be identified in the species with incomplete genomic data, yet these results likely demonstrate the ancestral adaptation of the late ISC pathway in Metamonada that involved the loss of Iba57 and IscA1 proteins [62].

## Discussion

This study presents an initial characterization of the late ISC pathway in anaerobic protist *G. intestinalis*. It shows an unexpected presence of BolA1 in the mitosomes, its interaction with Grx5 and the integration into the mitosomal interactome. Strikingly, mitosomes seem to lack actual client proteins that would require assembly of [4Fe-4S] cluster by the late ISC pathway and hence the actual function of *Gi*BolA1 and the entire late ISC pathway in the mitosomes remains unknown.

The independent evolution of mitochondria in various anaerobic lineages of eukaryotes resulted into remarkably uniform metabolic adaptations. Comparative studies on mitochondria and various MROs have suggested that the mitochondrial formation of Fe-S clusters was the main selection pressure for retaining the organelles even in the anoxic environments [5,49,63–66]. Mitochondria initiate the biosynthesis of cellular Fe-S clusters via the action of early ISC components that results into the formation of [2Fe-2S] cluster bound by glutaredoxin (Grx5) dimer. From here, the cluster is either distributed to mitochondrial clients, combined via late ISC components to [4Fe-4S] clusters or exported as an unknown sulfur-

containing factor to the cytosol [64]. Most of mitochondrial Fe-S client proteins contain [4Fe-4S] clusters and thus the late ISC pathway is vital for the function of the respiratory chain, the TCA cycle as well as the synthesis of prosthetic groups such as heme, lipoic acid or molybdenum cofactor [64]. Number of late ISC components are dedicated to serve these multiple clients in mitochondria and some of them could be also identified in *G. intestinalis*.

In this study, we show that despite the loss of all mitochondrial pathways that require the presence of [4Fe-4S] clusters, mitosomes of *G. intestinalis* retain four late ISC components; Grx5, IscA2, Nfu1 and the newly identified BolA1 homologue. In classical experimental models of yeast and mammalian mitochondria, defective late ISC pathway is often lethal for the cell or at least lead to severe diseases in humans due to multifactorial deficiencies caused in the mitochondrial metabolism [43,67]. In this context, mitosomes represent a unique biological model to study the non-mitochondrial role of the ISC pathway without the interference with mitochondrial metabolism.

Eukaryotes have three BolA proteins that function together with glutaredoxins in chaperoning the Fe-S cluster both in cytosol and mitochondria [25,27,68]. Yet, the previously reported absence of BolA proteins in the anaerobic eukaryotes that carry MROs suggested that BolA proteins are involved in the aerobic metabolism by controlling thiol redox potential [28]. Mitochondrial BolA1 and BolA3 were proposed to function as [4Fe-4S] assembly cluster factors via the interaction with Grx5 and Nfu1, respectively [25,27,69]. While the BolA3-Nfu1 interaction is required for the final [4Fe-4S] cluster transfer to the apoprotein [25], the exact role of BolA1-Grx5 in the preceding steps remains rather unknown. *Gi*BolA1 specifically interacts with *Gi*Grx5 as demonstrated by Y2H assay and the pulldown experiment. The interaction of *Gi*BolA1 with *Gi*Nfu1 was not supported by Y2H assay, yet the *Gi*Nfu1 was among the most enriched proteins co-purified with *Gi*BolA1. These data indirectly support the results of the phylogenetic reconstructions assigning *Gi*BolA1 to BolA1 proteins. Interestingly, for yet unknown reason the Grx5-BolA1pair is expendable in some Metamonada species, some of which carry metabolically versatile ATP-producing MROs, *e.g.*, the parabasalids and anaeramoebids.

Based on this data and the existence of BolA deficient yeast cell lines that exhibited relatively mild phenotype [27], the gene was selected for the targeted removal from *G. intestinalis* genome by CRISPR/Cas9. The assumption was that the gene would not be essential for *G. intestinalis* either. In addition, such a viable mutant could also reveal the general function of mitosomes in Fe-S cluster formation. Indeed, removal of the gene encoding *Gi*BolA1 by CRISPR/Cas9 showed that this protein is not essential for *G. intestinalis* maintained under laboratory conditions. In fact, ΔbolA1 cell line showed a no defect in mitosomal biology, specifically the mitosome number or the ability to form the [2Fe-2S] clusters during the *in vitro* assay. The first is consistent with the previous observation that *G. intestinalis* does not respond to metabolic perturbations by altering the number or distribution of mitosomes [70]. The latter finding supports the idea that *Gi*BolA1 is not required for the early ISC pathway by two means. It was not required during the *in vitro* formation of [2Fe-2S] cluster on experimental apoprotein and also its absence in the mitosomes did not affect the assembly of [2Fe-2S] cluster on the endogenous mitosomal ferredoxin that is required to provide electrons during the reductive steps of cluster assembly.

In contrast to the aerobic mitochondria [52,53], the removal of *Gi*BolA1 or *Gi*Gxr5 did not show any defects in the iron metabolism as demonstrated by its unchanged incorporation to target proteins. The key question remains, what is the function of *Gi*BolA1, i.e. what are the clients of mitosomal late ISC pathway? In yeast and patient-derived cell lines, BolA deficiency is manifested by a decrease in the activity of the [4Fe-4S] cluster containing protein succinate dehydrogenase, but also of pyruvate and 2-ketoglutarate dehydrogenases due to impaired

lipoylation by [4Fe-4S] lipoate synthase [27,38]. Since *G. intestinalis* does not encode any of these proteins, we searched in its genome for yet unknown mitosomal [4Fe-4S] proteins. We also analyzed the interactome of *Gi*Nfu1 as the experiments from other cellular model reported its co-purification with the client proteins [25,71,72]. None of the approaches led to the identification of such possible substrate but we cannot completely rule out the possibility that some yet unknown mitosomal client, that fell through the sieve of these analyses, requires the activity of *Gi*BolA, *Gi*Grx5 and other two mitosomal late ISC components. Alternatively, these proteins may participate in the formation of a cluster intermediate for nonmitosomal Fe-S proteins thereby providing link between the early ISC pathway and the cytosolic CIA pathway.

Interestingly, the function of Grx5 and BolA1 may be stage-specific in *G. intestinalis* as their genes together with IscU are significantly upregulated at the cyst stage [73]. The transcription profile of these genes is quite unusual as the genes are upregulated specifically in the cyst and not in the preceding encysting stages as it is observed for proteins involved in the cyst formation [73]. The expression of other ISC components remains either unaffected (ferredoxin, IscA2) or downregulated (IscS, Nfu1), hereby indicating a specific active role of *Gi*Grx5 and *Gi*BolA1 in the dormant infectious stage of the parasite.

In mitochondria, the Atm1 transporter in the inner membrane was shown to link the early ISC pathway with the cytosolic iron–sulphur assembly (CIA) via the transport of an unknown sulfur-containing molecule [74]. Atm1 homologue is missing in *G. intestinalis* and so are any other metabolic transporters or carriers. Interestingly, the interactome of the late ISC components presented in this work indicated that *Gi*IscA2 may play role in linking the mitosomal ISC and the cytosolic CIA machinery. The specific interaction of *Gi*IscA2 with the proteins in the outer mitosomal membrane and the sensitivity to the outer membrane solubilization indicated that it may in fact reside, at least partially, in the IMS of the mitosomes. Such localization would represent a unique adaptation of *G. intestinalis* mitosomes. Interestingly, several *G. intestinalis* CIA components, Cia2 and two Nbp35, were found to be partly localized or associated with the mitosomes [48], so it is tempting to speculate that the two pathways meet at the periphery of the mitosomes to hand over the cluster or its intermediate.

Of course, further experiments are needed to describe the actual function and submitosomal localization of *Gi*IscA2 that may represent a key factor for the mitosomal late ISC pathway and its impact on nonmitosomal Fe-S cluster assembly. To conclude, this work shows how late ISC pathway has undergone specific functional adaptations in a eukaryote inhabiting anoxic environments.

## Materials and methods

### Bioinformatics

The structural models of human and *G. intestinalis* BolA1 were computed using the Google Colabinterface of AlphaFold2 [75]. The multiple sequence alignment was generated with the jackhmmer option. The best scoring structure according to the plDDT score was subsequently refined with the Amber-Relax option. The [Fe-S] proteins were predicted by Metalpredator [47] using the conceptual proteome of *G. intestinalis* WBc6 strain (giardiadb.org).

### Phylogenetic dataset construction and inferences

Human BolA proteins (NP_001307954.1, NP_001307536.2, NP_997717.2) and *Giardia intestinalis* BolA1 were used as a query against NCBI non-redundant (nr) database to retrieve sequences from select Opishthokonta (*Danio renio, Mus musculus, Caenorhabditis elegans Schizosaccharomyces pombe Saccharomyces cerevisiae*), select Viridiplantae (*Glycine max Arabidopsis thaliana Chlamydomonas reinhardtii Chlorella variabilis*) and non-opisthokonts and

non-Viridiplantae (by restricting the database to non-opisthokonts and non-Viridiplantae) with an e-value threshold of 1e$^{-3}$. We also examined the predicted proteomes of metamonads available on EukProt v2 [76] and various sequencing initiatives [30,77,78]. The resulting queries were clustered based on sequence identity whereby using cd-hit [79] with a cut-off value of 0.9. Sequences were aligned using mafft (—auto) [80] and ambiguously aligned positions were removed using trimal with '-gt 0.5' [81]. Phylogenetic inference was performed using IQTREE2 to generate 1000 ultrafast bootstraps (-bb 1000) [82] under the LG+C60+G model of evolution (computed using -mset LG+C20,LG+C10,LG+C60,LG+C30,LG+C40,LG+C50, LG). Trees were visualized using FigTree v1.4 and stylized in Adobe Illustrator. Alignments and tree files are available at figshare (http://doi.org/10.6084/m9.figshare.19772155).

## Cloning and protein expression

For the expression of BAP-tagged proteins in *G. intestinalis*, the genes were amplified from genomic DNA and inserted into to pONDRA plasmid encoding the C-terminal BAP tag [83]. All the primers and the restriction enzymes used in this study are listed in S5 Table. Transfection was done as previously described [84] For the *in vivo* biotinylation, the cells expressing BAP-tagged proteins were transfected with a pTG plasmid encoding cytosolic BirA gene from *E. coli* [3]. For Y2H assay, genes were amplified from gDNA and subcloned to both pGADT7 and pGBKT7 plasmids. Mutated versions of genes for Y2H assay were commercially synthesized (Genscript).

For CRISPR/Cas9-mediated knockout of *bola1* gene, gRNA sequence ATCAGCTCTCCC GACTTCAA was inserted into gRNA cassette of pTGuide vector using [42] two annealed oligonucleotides (see S5 Table for primers and restriction enzymes used). The 999 bp of 5′ and 940 bp 3′ homologous arms surrounding *bolA* gene were inserted into pTGuide vector as the homologous arms for the recombination of the resistance cassette (S5 Table).

For CRISPR/Cas9-mediated knockout of *grx5* gene, four multiplexed gRNAs (CTAAGGC ACTTGCACTGACG, TTGGTAGGAATAGGGATCTG, AATGCCAGTGTGTGCTCCCT and GATGGATTCGCTCTGCGCCA) were inserted into 4 consecutive gRNA cassettes within the pTGuide vector [42] using two annealed oligonucleotides (see S5 Table for primers and restriction enzymes used). The 526 bp of 5′ and 608 bp 3′ homologous arms surrounding *grx5* gene were inserted into pTGuide vector as the homologous arms for the recombination of the resistance cassette (S5 Table).

## Cell culture, fractionation and immunoblot analysis

Trophozoites of *G. intestinalis* strain WB (ATCC 30957) were grown in TYI-S-33 medium [85] supplemented with 10% heat-inactivated bovine serum (PAA laboratories), 0,1% bovine bile and antibiotics. Cells were harvested and fractionated as previously described [3]. Cells expressing BAP-tagged *Gi*BolA1, *Gi*Grx5, *Gi*Nfu1, and *Gi*IscA2 were harvested and fractionated as previously described [3] Briefly, the cells were harvested in ice cold phosphate buffered saline (PBS, pH 7.4) by centrifugation at $1,000 \times g$, 4°C for 10 min, washed in SM buffer (20 mM MOPS, 250 mM sucrose, pH 7.4), and collected by centrifugation. Cell pellets were resuspended in SM buffer supplemented with protease inhibitors (Roche). Cells were lysed on ice by sonication for 2 min (1 s pulses, 40% amplitude). The lysate was centrifuged at $2,680 \times g$, for 20 min at 4°C to sediment the nuclei, cytoskeleton, and remaining unbroken cells. The supernatant was centrifuged at $180,000 \times g$, for 30 min at 4°C. The resulting supernatant corresponded to the cytosolic fraction, and the high-speed pellet (HSP) contained organelles including the mitosomes and the endoplasmic reticulum. The *Gi*Nfu1, *Gi*IscA2, *Gi*Grx5 and *Gi*BolA1 proteins were detected by a rabbit anti-BAP polyclonal antibody (GenScript).

Mitosomal *Gi*Tom40 and *Gi*IscU were detected with a specific polyclonal antibody raised in rabbits [59]. The primary antibodies were recognized by secondary antibodies conjugated with horseradish peroxidase. The signals were visualized by chemiluminescence using an Amersham Imager 600.

## Immunofluorescence microscopy

*G. intestinalis* trophozoites were fixed and immunolabeled as previously described [70,86]. The C-terminal BAP tag of localized mitosomal proteins was detected by a rabbit anti-BAP polyclonal antibody (GenScript). Mitosomal marker GL50803_9296 was detected by a rabbit anti- GL50803_9296 polyclonal antibody [3]. The primary antibodies were detected by secondary antibodies included: Alexa Fluor 594 donkey anti-rabbit IgG (Invitrogen), Alexa Fluor 488 donkey anti-mouse IgG (Invitrogen). Slides were mounted in Vectashield containing DAPI (Vector Laboratories).

Static images were acquired on Leica SP8 FLIM inverted confocal microscope equipped with 405 nm and white light (470–670 nm) lasers and FOV SP8 scanner using HC PL APO CS2 63x/1.4 NA oil-immersion objective. Laser wavelengths and intensities were controlled by a combination of AOTF (Acousto-Optical Tunable Filter) and AOBS (Acousto-Optical Beam Splitter) separately for each channel. Emitting fluorescence was captured by internal spectrally-tunable HyD detectors. Imaging was controlled by the Leica LAS-X software. Images were deconvolved using SVI Huygens software with the CMLE algorithm. Maximum intensity projections and brightness/contrast corrections were performed in FIJI ImageJ software [87].

## *In vitro* Fe-S cluster reconstitution assay

[2Fe-2S] Ferredoxin type 4 from *T. vaginalis* was expressed and purified from *E. coli* as reported earlier [49]. To prepare apo-ferredoxin, the purified holo-ferredoxin was incubated with 0.5M HCl for 10 minutes on ice and the sample was then neutralized bythe addition of Tris (pH 7.5) to a final concentration of 0.6 M. Dissociated [Fe-S] clusters and other small molecules were removed using 7K MWCO Zeba Spin Desalting columns (Thermo Scientific) according to the manufacturer's instructions. For the *in vitro* assay, organellar (HSP) and cytosolic fractions of *G. intestinalis* were prepared as described above. The reactions were carried out by mixing total organellar or cytosolic fraction and apo-ferredoxin in a 4: 1 ratio (w/w) in a buffer containing 0.5% Triton X-100, 50 μM ferrous ascorbate, 20 mM HEPES (pH 8.0), 25 μM L-cysteine and 10 μCi of [$^{35}$S]-L-cysteine. The mixture was incubated at 25˚C for 60 min and subsequently stopped by 5 mM EDTA. Unincorporated [$^{35}$S]-L-cysteine and other small molecules were separated from the reconstituted holo-ferredoxin using 7K MWCO Zeba Spin Desalting columns (Thermo Scientific). The samples were separated on 15% non-denaturing polyacrylamide gel at 4˚C. The gels were vacuum dried for 2 hours, autoradiographed using a BAS-IP TR 2025 E tritium storage phosphor screen (GE Healthcare) and visualized by Typhoon FLA 7000 (GE Healthcare).

## Iron uptake and incorporation to *G. intestinalis* proteins

The assay was performed as recently published [88]. Briefly, trophozoites of *G. intestinalis* expressing Cas9 (parental cell line), Δ*bolA1* and Δ*grx5* were grown in TYI-S-33 culture media supplemented with 0.5μM $^{55}$Fe (29,600 MBq mg$^{-1}$) in the form of ferric citrate (1:20) and incubated for 72 hours. After incubation, the cells were harvested by centrifugation and washed three times with NaCl-HEPES buffer (0.14M NaCl, 10mM HEPES, pH 7.4). Cells were disrupted by sonication in the NaCl-HEPES buffer containing 1% digitonin and cOmplete EDTA-free protease inhibitor cocktail (Roche). Protein concentration was assessed using a

BCA kit (Sigma-Aldrich), and an equal concentration of proteins was separated using Novex Native PAGE Bis–Tris Gel system (4–16%; Invitrogen) according to the manufacturer's protocol. Gels were vacuum-dried and autoradiographed for 7 days using a BAS-IP TR 2025 E tritium storage phosphor screen (GE Healthcare) and visualized by Typhoon FLA 7000 (GE Healthcare).

## Cross-linking, protein isolation, mass spectrometry (MS)

The HSP (10 mg) isolated from each cell line was collected by centrifugation (30 000 x g, 4˚C, 10 min) and resuspend in 1 x PBS supplemented with protease inhibitors (Roche) to protein concentration 1.5 mg/ml. The cross-linker DSP (dithiobis(succinimidyl propionate), Thermo-Scientific) was added to final 100 μM concentration. The sample was incubated 1 h on ice. Crosslinking was stopped by the addition of 50 mM Tris (pH 8.0) followed by 15 min incubation at RT. The sample was collected by centrifugation (30 000 x g, 10 min, RT) and then resuspended in boiling buffer (50 mM Tris, 1mM EDTA, 1% SDS, pH 7.4) supplemented with protease inhibitors. The sample was then incubated at 80˚C for 10 min, collected by centrifugation and the supernatant was diluted 1/10 in the incubation buffer (50 mM Tris, 150 mM NaCl, 5 mM EDTA, 1% Triton X-100, pH 7.4) supplemented with protease inhibitors. Streptavidin-coupled magnetic beads (50 μL of Dynabeads MyOne Streptavidin C1, Invitrogen) were washed three times in 1 ml of the incubation buffer for 5 min and added to the sample, mixed and incubated for 1 h at room temperature and then incubated overnight with gentle rotation at 4˚C. The beads with bound protein were washed three times in the incubation buffer (5 ml) supplemented with 0.1% SDS for 5 min, washed in boiling buffer for 5 min and then washed in the washing buffer (60 mM Tris, 2% SDS, 10% glycerol, 0.1% SDC) for 5 min. Finally, the sample was washed twice in 100 mM TEAB (Triethylammonium bicarbonate, Thermofisher) with 0,1% SDC for 5 min. One tenth of the sample was mixed with SDS-PAGE sample buffer supplemented with 20 mM biotin and incubated in 95˚C for 5 min. Experimental controls were tested by immunoblotting and then the sample (dry frozen beads with proteins) was analyzed by mass spectrometry. Control sample was processed in the same way. Each sample was done in triplicate. Beads with bound proteins were submitted to tandem mass spectrometry (MS/MS) analysis as previously described except without the detergent washing steps [84]. In brief, captured samples were released from beads by trypsin cleavage. Peptides were separated by reverse phase liquid chromatography and eluted peptides were converted to gas-phase ions by electrospray and analyzed using an Orbitrap (Thermo Scientific, Waltham, MA) followed by Tandem MS to fragment the peptides through a quadropole for final mass detection. Data was analyzed using MaxQuant (version 1.6.3.4) [89] with a false discovery rate (FDR) of 1% for both proteins and peptides and a minimum peptide length of seven amino acids. The Andromeda search engine [90] was used for the MS/MS spectra search against the latest version of the *G. intestinalis* database from EuPathDb (http://eupathdb.org/eupathdb/) and a common contaminant database. Modifications were set as follows: Cystein (unimod nr: 39) as static, and methionoine oxidation (unimod: 1384) and protein N terminus acetylation (unimod: 1) as variable. Data analyses were performed using Perseus 1.6.1.3 [91] and visualized as a volcano plot using the online tool VolcaNoseR (fold change 1,significance threshold 2) [92] and as a heatmap using the online tool ClustVis [93].

## Protease protection and digitonin solubilization assays

For protease protection assay, cells expressing BAP-tagged *Gi*BolA, *Gi*Grx5, *Gi*Nfu1, and *Gi*IscA2 were harvested and fractionated as described above. The HSP fraction (150 μg) was resuspended in 20 μl of SM buffer and supplemented with protease inhibitors, or 20 μg/ml of

trypsin or 20 μg/ml of trypsin and 0.1% Triton X-100. The samples were incubated 30 min at 25˚C and then processed for SDS-PAGE.

For digitonin solubilization assay, 100 μg of HSP fractions isolated from cells co-expressing HA-tagged *Gi*IscU and BAP-tagged *Gi*IscA2 were incubated for 30 min on ice with 0.01%, 0.05%, 0.1%, digitonin, and without digitonin as a control. The samples were diluted by PBS to 800 μl total volume and collected by centrifugation (30 mins, 180,000 × g, at 4˚ C). The resulting pellets were processed for SDS-PAGE and the supernatants were precipitated by 15% TCA for 30 min on ice and collected by centrifugation for 30 min at 180,000 × g and 4˚ C, the pellets were washed once with 500 μl of ice-cold acetone, centrifuged as before. The samples were resolved by SDS-PAGE, transferred to nitrocellulose membrane and the protein tags were detected by rabbit anti-BAP antibody (Genscript) and rat anti-HA antibody (Roche). The release to mitosomal proteins was quantified by ImageJ [87].

### Y2H assay

The yeast two-hybrid assay (Y2H) was performed as previously described [94]. *S. cerevisiae* cells (strain AH109) were co-transformed with two plasmids (pGADT7, pGBKT7) with the following combinations of genes: *Gi*BolA1 + *Gi*Grx5, *Gi*BolA1 + *Gi*mGrx5 (C128A-mutated Grx5), *Gi*Grx5 + *Gi*mBolA1 (H90A-mutated *Gi*BolA1). The empty plasmids were used as negative controls. Co-transformants were selected on double dropout plates SD -Leu/-Trp and triple dropout plates SD -Leu/-Trp/-His. The colonies were grown for four days at 30˚C. The positive colonies from triple dropout medium were grown overnight at 30˚C, 200 RPM and then the serial dilution test was performed on double and triple dropout plates.

## Supporting information

**S1 Fig. Protein sequence alignments of late ISC components of *Giardia intestinalis*.** (A) Grx5, the diagram shows the domain structure of *Gi*Grx5, mitochondrial targeting sequence (MTS) is shown in red, monothiol glutaredoxin domain (PF00462) in purple, the CGFS motif is also highlighted. (B) *Gi*IscA2 shares the Fe-S_biosyn domain (PF01521) with the conserved cysteine residues involved in cluster binding. (C) *Gi*Nfu1 contains conserved N- and C-domains, the latter of is recognized as NifU domain (PF01106) and carries conserved cysteine motif.
(PDF)

**S2 Fig. Full blots of cellular fractions and protease protection assay experiments.**
(PDF)

**S3 Fig. Serial dilutions of Y2H assay testing the protein interactions between *Gi*BolA1 and *Gi*Nfu1.**
(PDF)

**S4 Fig. Mitosomal morphology and number is not affected by the removal of *bolA1* gene.** The exemplary image of mitosomes visualized by immunofluorescence microscopy in the ΔbolA1 and control (Cas9) cell lines. Mitosomes were detected by rabbit polyclonal antibody raised against GL50803_9296, the nuclei were stained with DAPI.
(PDF)

**S5 Fig. Establishment of Grx5 knockout (Δgrx5) cell line.** The Δgrx5 cell line was tested for the presence of *grx5* gene and the integration of homologous recombination cassette (HRC) by PCR on gDNA, (B) the expression of *grx5* gene in Δgrx5 cell line was tested by PCR on the cDNA, *β-giardin* was used as a control gene, (C) Incorporation of $^{55}$Fe to *G. intestinalis*

proteins after 72 h incubation with radioactive iron isotope in the form of ferric citrate. Comparisons of control and Δgrx5 cell extracts show comparable levels of iron incorporation. (PDF)

**S1 Table. Proteomic analysis of *Gi*BolA, *Gi*Grx5, *Gi*Nfu1 and *Gi*IscA2 pulldowns.** For all proteins, statistical analysis based upon the biological and technical triplicates are shown. (XLSX)

**S2 Table. Fe-S proteins of *G. intestinalis*.** (XLSX)

**S3 Table. Proteomic analysis of ΔbolA1 cell line.** (XLSX)

**S4 Table. ISC components of Metamonada** (XLSX)

**S5 Table. Primers used in the study.** (XLSX)

## Author Contributions

**Conceptualization:** Alžběta Motyčková, Luboš Voleman, Staffan Svärd, Courtney W. Stairs, Pavel Doležal.

**Data curation:** Lenka Arbonová, Vít Dohnálek, Natalia Janowicz, Ronald Malych, Róbert Šuťák, Courtney W. Stairs, Pavel Doležal.

**Formal analysis:** Alžběta Motyčková, Luboš Voleman, Vladimíra Najdrová, Lenka Arbonová, Martin Benda, Vít Dohnálek, Ronald Malych, Róbert Šuťák, Courtney W. Stairs, Pavel Doležal.

**Funding acquisition:** Thijs J. G. Ettema, Staffan Svärd, Courtney W. Stairs, Pavel Doležal.

**Investigation:** Alžběta Motyčková, Natalia Janowicz, Courtney W. Stairs.

**Methodology:** Vladimíra Najdrová, Martin Benda, Courtney W. Stairs, Pavel Doležal.

**Resources:** Luboš Voleman.

**Supervision:** Pavel Doležal.

**Writing – original draft:** Alžběta Motyčková, Courtney W. Stairs, Pavel Doležal.

**Writing – review & editing:** Thijs J. G. Ettema, Staffan Svärd, Pavel Doležal.

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
