## [Decision Letter · Decision Letter 0]

9 Sep 2022

Dear Dr. Dolezal,

Thank you very much for submitting your manuscript "The late ISC pathway interactome reveals mitosomal-cytoplasmic crosstalk in Giardia intestinalis" for consideration at PLOS Pathogens. As with all papers reviewed by the journal, your manuscript was reviewed by members of the editorial board and by several independent reviewers. In light of the reviews (below this email), we would like to invite the resubmission of a significantly-revised version that takes into account the reviewers' comments.

While two reviewers gave positive evaluation on your manuscript, the third reviewer evaluated it very critically. As Handling Editor, I am more toward the view of Reviewer 3, in particularly in that (1) the functionality of the enigmatic cytoplasmic ISC machinery is not further addressed in the current manuscript, while an obvious mitosomal 4Fe-4S target is still not yet identified; (2) the different behavior of Isa2 in this analysis method remains only descriptive (no functional study conducted); (3) functional complementation after KO was not conducted; plus mitosomal morphology and fusion/fission were not analyzed in KO. Therefore, the protein-protein interaction of Giardia Bol1 and the other late ISC proteins are significant and highly appreciable, functional data are largely missing and thus, the conclusions are often speculative. It would be convincing if the authors show in revision that knockouts of the other late ISC factors give the comparable outcomes to PFOR. It is probably nice, if not mandatory, if the authors show the evidence for the Grx5-Bol1 complex helping Fdx1 maturation and hence boll1 KO causing an early ISC defect. If you can address those points experimentally, I am happy to receive the revised manuscript that should have functional data, as explained above, to directly demonstrate a crosstalk between mitosomes and the cytoplasm. Please note that once the revision is received, it will be most likely sent out for review to the third referee.

We cannot make any decision about publication until we have seen the revised manuscript and your response to the reviewers' comments. Your revised manuscript is also likely to be sent to reviewers for further evaluation.

Sincerely,

Tomoyoshi Nozaki, M.D., Ph.D.

Associate Editor

PLOS Pathogens

Vern Carruthers

Section Editor

PLOS Pathogens

Kasturi Haldar

Editor-in-Chief

PLOS Pathogens

orcid.org/0000-0001-5065-158X

Michael Malim

Editor-in-Chief

PLOS Pathogens

orcid.org/0000-0002-7699-2064

While two reviewers gave positive evaluation on your manuscript, the third reviewer evaluated it very critically. As Handling Editor, I am more toward the view of Reviewer 3, in particularly in that (1) the functionality of the enigmatic cytoplasmic ISC machinery is not further addressed in the current manuscript, while an obvious mitosomal 4Fe-4S target is still not yet identified; (2) the different behavior of Isa2 in this analysis method remains only descriptive (no functional study conducted); (3) functional complementation after KO was not conducted; plus mitosomal morphology and fusion/fission were not analyzed in KO. Therefore, the protein-protein interaction of Giardia Bol1 and the other late ISC proteins are significant and highly appreciable, functional data are largely missing and thus, the conclusions are often speculative. It would be convincing if the authors show in revision that knockouts of the other late ISC factors give the comparable outcomes to PFOR. It is probably nice, if not mandatory, if the authors show the evidence for the Grx5-Bol1 complex helping Fdx1 maturation and hence boll1 KO causing an early ISC defect. If you can address those points experimentally, I am happy to receive the revised manuscript that should have functional data, as explained above, to directly demonstrate a crosstalk between mitosomes and the cytoplasm. Please note that once the revision is received, it will be most likely sent out for review to the third referee.

Reviewer's Responses to Questions

**Part I - Summary**

Reviewer #1: Mitosome in G. intestinalis is one of the simplest mitochondria-related organelles (MROs) and possesses iron-sulfur (Fe-S) cluster synthesis (ISC) pathway. In this study, the authors identified bolA in an anaerobic protist, G. intestinalis, which was reported to be absent from anaerobic eukaryotes. They showed GibolA is localized in mitosomes and mitosomal GibolA is important for assembly of [4F-4S] cluster, which is essential for cytosolic enzyme, PFOR. Biochemical characterization of ISC pathway in highly degenerated MROs is important in mitochondrial evolutional study, and this study addressing this issue is potentially very interesting. However, their conclusions are sometimes too strong to be accepted as is.

Reviewer #2: Authors performed interactome analysis of the late iron sulfur cluster biosynthesis pathway in Giardia intestinalis mitosomes. It led to an interesting discovery of an aerobic mitochondrial protein homolog BolA involved in this mitosomal process. Microscopy and biochemical analyses indicated to BolA lozalization to Gi mitosomes. Validation of the interactome analysis by Y2H was performed satisfactorily. Authors also reported the apparent absence of a client protein for the synthesized [4Fe-4S] cluster in mitosomes. Instead, authors link this process to the cytosolic assembly of [4Fe-4S] clusters by knockout of the bolA gene. Validation of the interactome analysis by Y2H was performed. These findings contribute to the understanding of ISC synthesis in non-model organisms particularly among metamonads. Key questions although not currently answerable experimentally, were somehow addressed by speculation or hypotheses from the authors.

Reviewer #3: This short manuscript addresses the existence, and to some extent function, of the so-called late ISC proteins in Giardia intestinalis. Normally, mitosome-containing species like Giardia possess only the early part of the ISC machinery, and lack the late one because these species do not contain any known 4Fe-4S proteins, the targets of the late ISC system. The authors (re )identify the already characterized Grx5 and the genes for Isa2 and Nfu1 in the genome of Giardia. In a newly released genome assembly of Giardia they now also find Bol1, a protein which in mitochondrial FeS protein assembly at best has an auxiliary function. Ind1 and Iba57 (and presumably Isa1) was not found. The existence of so many Giardia late ISC proteins is surprising because to date no 4Fe-4S targets in Giardia mitosomes are known. The authors then concentrate mostly on Bol1 and show that a tagged overexpressed protein is located in membrane fractions of cell extracts, and localizes to particulate structures that are known to represent mitosomes. The protein interacts with many mitosomal proteins including the other late (and also early) ISC proteins. Vice versa, the tagged overexpressed late ISC proteins Nfu1, Isa2, and Grx5 interact with a similar set of proteins, as shown by affinity pulldowns. Interestingly, native Bol1 was not reliably identified in the mass spec data sets, and the authors suggest low abundance of this protein. Nevertheless, the proteomic analysis suggests an interaction of these late mitosomal ISC factors, but the functional issue is not further addressed in the current manuscript, simply because an obvious mitosomal 4Fe-4S target is missing. Also, the slightly different behavior of Isa2 in this analysis method was not further analyzed and thus remains descriptive (see point 7 below). The authors then generate a delta-bol1 cell using the now available CRISPR method for Giardia (unfortunately without confirming the specificity of the KO by complementation). The knockout cells grew slower, yet the number (and morphology? see point 9 below) of mitosomes remained unchanged. Finally, and a bit out of the blue, the authors check the activity of PFOR, a cytosolic 4Fe-4S protein. Why this one? Its activity is clearly down in Bol1 knockout cells, but a convincing explanation why this happened is not provided. Also, controls are missing. This effect could be, as claimed, a role of Bol1 for cytosolic FeS maturation, or (at this stage equally likely) a completely indirect consequence of Bol1 deletion. Overall, the interaction findings on Giardia Bol1 and the other late ISC proteins are nice and confirm findings in many other organisms. Functionally, the data is quite preliminary and puzzling at this stage. Would knockouts of the other late ISC factors give the same result for PFOR? Is the effect due to the Grx5-Bol1 complex helping Fdx1 maturation and hence causing an early ISC defect? Many questions are unanswered that are central for the claim of a crosstalk between mitosomes and cytosol.

**Part II – Major Issues: Key Experiments Required for Acceptance**

Reviewer #1: Major Point

1. Line 29. The authors state, “specific interaction of IscA with the outer mitosomal membrane”. I found IscA interactome data that suggests the interaction of IscA with MOMP35, but I could not find the data for the interaction with membrane.

2. MetalPredator is a publicly available software to predict iron-sulfur cluster binding proteins in protein sequence databases; therefore, as the authors state, “Of course, we cannot rule out the presence of a previously unknown protein with a unique cluster binding domain/motif in mitosomes.”, it is highly possible to overlook. However, the author’s conclusions regarding iron-sulfur cluster binding proteins are too strong; for example, line 300, “Given the apparent absence of the client proteins in the mitosomes”. Line 334, “despite the loss of all mitochondrial pathways that require the presence of [4Fe-4S] clusters,”

3. Figure 1. It is OK as an introductory schematic figure, but the authors insert G. intestinalis enzymes information. However, the information included is so incomplete that readers may have difficulties to follow. It would be better to include authors’ conclusions drawn in this study such as GiBolA function, GiIscA localization, and the assembly of cytosolic [4F-4S] clusters.

4. Figure 3A-C. How many times did authors repeat the cross-linking, protein isolation, and mass spectrometry analysis? Is the enrichment of MOMP35 in IscA sample statistically significant?

5. Line 523. The authors state, “each sample was done in triplicate”. How variability in each sample is reflected in Volcano plots?

6. Line 166. Please include the data for “the analogous assay did not show any interaction between GiBolA and GiNfu1” in supporting information.

7. The bolA gene knockout strain was able to be obtained and showed normal growth until 2 days, indicating that bolA gene is not essential. How does this strain acquire [4Fe-4S] for PFOR?

Reviewer #2: (No Response)

Reviewer #3: Other comments:

1. Lanes 45,46: This half sentence sounds as if mitochondria have more than one pathway for FeS synthesis. What the authors mean is that there are other pathways in addition to the FeS. Please rephrase.

2. Lanes 52 onwards: It is confusing that the authors use the bacterial nomenclature for most ISC proteins. For instance, IscS in fact is Nfs1 or NFS1. Calling the mitochondrial-mitosomal proteins IscS etc. is incorrect. I therefore recommended using either the yeast or human names for these proteins.

3. Lanes 90-92: The statement “that mitosomal BolA, and thus the late ISC pathway, is required for the formation of cytosolic 4Fe-4S clusters.” is incorrect in that the functionality of the late ISC pathway was not touched upon in this work, just Bol1 was looked at.

4. Lane 98 ff.: Define in Suppl. Fig. 1B that it is NOT Isa1 (ISCA1) but Isa2 (ISCA2) in the alignment. The latter proteins from yeast and humans are used. Hence, use IscA2 rather than IscA (or even better, the name as recommended in point 2).

5. Fig. 2F is unclear to me. What do the authors mean by “absent from the transcriptome”? Gene present, but not transcribed?

6. Lane 131: The headline “GiBolA is part of mitosomal late ISC pathway” is an exaggeration. What was described in the respective chapter is the interaction of Bol1 with Giardia proteins, mostly located in mitosomes, not a pathway. The authors may therefore wanna be a bit more accurate in phrasing the headline.

7. Lanes 257-258: This statement is a bit of an exaggeration at this stage. This behavior may simply be a different membrane sticking of Isu1 and Isa2 proteins. Without further proof (e.g., salt variations; affinity purifications of Isu1, etc.) I suggest to tone down the statement.

8. Lanes 289-291: Rephrase the sentence, please. It seems incomplete (or the two parts are not connected properly?).

9. Lane 298: Typo: Suppl. Fig. 3 instead of 2. More important: How did the authors define “morphology”? Simply by an unchanged particulate appearance in IF? If yes, this should be stated explicitly, because it would not really reflect morphology (which in my view would need EM analysis).

**Part III – Minor Issues: Editorial and Data Presentation Modifications**

Reviewer #1: Minor Point

1. Figure 1E. Which is GiBolA?

2. Line 306 and Materials and Methods. There is no description how the PFOR activity was assayed.

3. Figure 1D. Most readers will be misled abbreviations ‘mGiGrx5’ and ‘mGiBolA’ to ‘mitosomal’ or ‘mitochondrial’. It would be easier for readers to abbreviate usual form, such as GiGrx5(C128A) (or GiGrx5C128A) and GiBolA(H90A) (or GiBolAH90A).

4. Line 136. Similar to 9, readers will confuse high-speed pellet (HSP) with heat-shock protein like Hsp70.

5. Line 382. The authors should consider removing (Grx??? Is this known?)

Reviewer #2: Please describe the mitosomal marker GL50803_9296 in line 135.

Edit line 382.

Review the entire text for scientific names are not italicized.

Reviewer #3: (No Response)

PLOS authors have the option to publish the peer review history of their article (what does this mean?). If published, this will include your full peer review and any attached files.

Reviewer #1: No

Reviewer #2: No

Reviewer #3: No
---

## [Decision Letter · Decision Letter 1]

17 Sep 2023

Dear Dr. Dolezal,

We are pleased to inform you that your manuscript 'Adaptation of the late ISC pathway in the anaerobic mitochondrial organelles of  Giardia intestinalis.' has been provisionally accepted for publication in PLOS Pathogens.

Best regards,

Tomoyoshi Nozaki, M.D., Ph.D.

Academic Editor

PLOS Pathogens

Vern Carruthers

%CORR_ED_EDITOR_ROLE%

PLOS Pathogens

Kasturi Haldar

Editor-in-Chief

PLOS Pathogens

orcid.org/0000-0001-5065-158X

Michael Malim

Editor-in-Chief

PLOS Pathogens

orcid.org/0000-0002-7699-2064

Reviewer Comments (if any, and for reference):

Reviewer's Responses to Questions

**Part I - Summary**

Reviewer #1: The revised version of the manuscript has effectively addressed my primary concerns.

Reviewer #2: (No Response)

**Part II – Major Issues: Key Experiments Required for Acceptance**

Reviewer #1: (No Response)

Reviewer #2: (No Response)

**Part III – Minor Issues: Editorial and Data Presentation Modifications**

Reviewer #1: (No Response)

Reviewer #2: (No Response)

PLOS authors have the option to publish the peer review history of their article (what does this mean?). If published, this will include your full peer review and any attached files.

Reviewer #1: No

Reviewer #2: No

---

## [Editor Report · Acceptance letter]

28 Sep 2023

Dear Dr. Doležal,

We are delighted to inform you that your manuscript, "Adaptation of the late ISC pathway in the anaerobic mitochondrial organelles of *Giardia intestinalis*.," has been formally accepted for publication in PLOS Pathogens.

Best regards,

Kasturi Haldar

Editor-in-Chief

PLOS Pathogens

orcid.org/0000-0001-5065-158X

Michael Malim

Editor-in-Chief

PLOS Pathogens

orcid.org/0000-0002-7699-2064